# Liver Progenitor Cells in Massive Hepatic Necrosis—How Can a Patient Survive Acute Liver Failure?

**DOI:** 10.3390/biom12010066

**Published:** 2022-01-02

**Authors:** Tao Lin, Rilu Feng, Roman Liebe, Hong-Lei Weng

**Affiliations:** 1Department of Medicine II, Division of Hepatology, University Medical Center Mannheim, Medical Faculty Mannheim, Heidelberg University, 68167 Mannheim, Germany; tao.lin@medma.uni-heidelberg.de (T.L.); rilu.feng@medma.uni-heidelberg.de (R.F.); 2Clinic of Gastroenterology, Hepatology and Infectious Diseases, Heinrich Heine University, 40225 Düsseldorf, Germany; r.mullenbach@gmail.com

**Keywords:** acute liver failure, acute-on-chronic liver failure, activin, coagulation factor, HNF4α, liver progenitor cell, massive hepatic necrosis, sepsis, systemic inflammatory response syndrome

## Abstract

Massive hepatic necrosis is the most severe lesion in acute liver failure, yet a portion of patients manage to survive and recover from this high-risk and harsh disease syndrome. The mechanisms underlying recovery remain largely unknown to date. Recent research progress highlights a key role of liver progenitor cells, the smallest biliary cells, in the maintenance of liver homeostasis and thus survival. These stem-like cells rapidly proliferate and take over crucial hepatocyte functions in a severely damaged liver. Hence, the new findings not only add to our understanding of the huge regenerative capability of the liver, but also provide potential new targets for the pharmacological management of acute liver failure in clinical practice.

## 1. Introduction

The current commonly accepted nomenclature of acute liver failure (ALF), with alternative terms including “fulminant hepatitis” and “fulminant hepatic failure”, was coined by Roger Williams and colleagues in 1993 [1]. Their ALF definition describes a severe liver injury leading to coagulation abnormality, usually with an international normalized ratio (INR) ≥1.5, and any degree of mental alteration (encephalopathy) in a patient without pre-existing liver disease and with an illness of up to 4 weeks of duration [1]. Historically, ALF was systemically scrutinized for the first time by Lucke and Mallory in 1940s [2]. Based on two autopsy studies of 178 soldiers who had died of viral hepatitis break during the Second World War, the authors confirmed massive hepatic necrosis (MHN) as the core histological characteristics of this “fulminant form of epidemic hepatitis” [2]. The following clinicopathological study showed that MHN is the most severe lesion in acute liver failure [3]. The most frequent etiologies of ALF are hepatitis A, B, and E virus infection and drug hepatotoxicity. In addition, ischemic/hypoxic injury, Wilson disease, and autoimmune hepatitis (AIH) in adults and metabolic diseases, hereditary tyrosinemia type 1, and AIH in children and infants also cause ALF [3]. Alcoholic hepatitis and hepatitis C virus also lead to encephalopathy and coagulation abnormality, but do not demonstrate MHN [3]. 

For a long time, ALF included patients with pre-existing liver disease, particularly those with chronic HBV infection [4]. Subsequently, the term acute-on-chronic liver failure (ACLF) emerged, describing an emergency syndrome that occurred mainly in cirrhotic patients with alcoholic hepatitis and HCV infection [5]. ALF was recommended to be limited to patients without pre-existing liver disease in order to distinguish ACLF [3]. Such a suggestion does not reduce the debate on how to distinguish acute decompensation between ALF, ACLF, European association for the Study of the Liver (EASL)-ACLF, sepsis and sepsis in cirrhosis. In this review, we discuss how patients can survive MHN, either with or without pre-existing liver disease. We focus on the mechanisms of how a severely damaged liver can maintain essential functions to support systemic homeostasis while undergoing such a severe and dangerous clinical syndrome. The definitions of ALF, ACLF, EASL-ACLF, sepsis and sepsis in cirrhosis, are also briefly discussed in an isolated section. 

## 2. Who Takes over Hepatocyte Functions to Support Systemic Homeostasis Following Massive Hepatic Necrosis?

As we discussed in a previous review [6], no consensus MHN definition is available to date. If the entire liver on explant or autopsy is available, MHN is defined as extensive, diffuse panlobular (panacinar) and multilobular necrosis of >60–70% of the liver [7]. This definition is elegantly illustrated in the patient cohorts presented by Lucke and Mallory [2]. In some ALF patients, it is difficult to find residual hepatocytes (Figures 8 and 9 in [2]). Impressively, most of these patients still survived at least several days after the onset of the acute decompensation. This raises an interesting question: how can these patients survive in the near absence of hepatocytes?

### 2.1. Liver Progenitor Cells Take over Hepatocyte Functions Following Massive Hepatic Necrosis

Hepatocytes perform multiple indispensable functions in systemic homeostasis, including the production of albumin, bile, and most coagulation factors, as well as metabolism of carbohydrates, proteins, fats, hormones, bilirubin, and ammonia [8]. Among these functions, the production of albumin, bile, and most coagulation factors and the metabolism of hormones, bilirubin, and ammonia completely depend on hepatocytes in physiological condition. It is noteworthy that hepatocyte-synthesized proteins have a wide range of half-lives: albumin remains in circulation for more than 20 days, whereas the half-life of coagulation factors is only several hours [8]. Without coagulation factors, patients cannot survive. In ALF patients suffering from HMN, there is a high demand of coagulation factors. This raises the question of how ALF patients can survive for days or even recover in the absence of most hepatocytes. Reviewing 178 ALF patients scrutinized by Lucke and Mallory [2], 94 patients demonstrated a paucity of hepatocytes in autopsied livers, indicating the occurrence of MHN. Nevertheless, these patients survived for between 1 and 9 days following the onset of acute decompensation. Impressively, even in the two ALF patients whose duration was only 1 day, proliferative ducts and active liver progenitor cells (LPC) could be detected [2]. For a long time, pathologists thought that the maintenance of vital hepatocyte functions in ALF patients following massive hepatic necrosis mainly depended on sufficient abundance and function of activated liver progenitor cells [6,9]. Our recent study provided evidence on how LPC produce coagulation factors following MHN [10]. 

Physiologically, the transcription of most coagulation factor genes is regulated by master hepatic transcription factors such as HNF4α [11]. In normal conditions, HNF4α is constitutively expressed by hepatocytes. No additional liver cells express this factor physiologically. However, LPC in ALF patients, particularly in recovered patients, robustly express HNF4α as well as coagulation factors. Impressively, in those recovered patients, strong nuclear HNF4α immune reactivity in LPC was only observed in areas without hepatocytes, whereas immune reactivity of HNF4α in LPC was very weak or negative in areas with remaining hepatocytes (Figure 1 in [10]), indicating that activated LPC take over hepatocyte functions only in the sections lacking hepatocytes. As in hepatocytes, HNF4α in LPC controls multiple coagulation factors by binding to their gene promoters [10]. These results provide an explanation how ALF patients possess capability to restore coagulation function in the condition of MHN.

### 2.2. How Do LPC Initiate Robust HNF4α Expression?

Normal LPCs reside in the canals of Hering and ductules. In normal livers, neither canals of Hering nor ductules are easily seen, but they may become apparent in the disease state [13]. With immunohistochemical staining for biliary markers, e.g., CK19, CK7 and SOX9, the canals of Hering can be identified and appear as isolated cuboidal or strings of cells [14]. HNF4α expression in normal LPC is undetectable. ALF patients who subsequently receive liver transplantation, defined as “irreversible ALF”, also do not express robust HNF4α in LPC. As mentioned above, even in the surviving ALF patients, HNF4α in LPC is undetectable in areas with ample numbers of surviving hepatocytes [10]. Only in the areas without remaining hepatocytes do active LPC express HNF4α [10]. These results imply that signals from the microenvironment surrounding LPC, such as dead hepatocytes or inflammatory cells, might play a crucial role in regulating HNF4α expression. Our recent study showed that activin A is a factor stimulating the expression of HNF4α in LPC [10]. As a member of the TGF-β superfamily, activin A plays a crucial role in embryonic development [15,16]. In the condition of inflammation, macrophages and dendritic cells can produce Activin A [17]. In the normal liver, hepatocytes are a major source of Activin A [18]. Massive hepatic necrosis might produce a large amount of Activin A, although it is difficult to determine serum Activin A concentrations at the exact time point when MHN occurs. Activin A induces HNF4α expression in LPC through a transcription factor complex formed by its downstream transcription factors SMAD2/3/4 and the cofactor FOXH1 [10]. Interestingly, serum activin concentrations in both the surviving and irreversible ALF patients measured before histological analyses were at similar levels. However, histological examination revealed that LPC in the surviving ALF patients showed robust p-SMAD2 immune positivity, whereas those in the irreversible patients did not. These results suggest that there is/are factor(s) inhibiting activin A signaling in LPC of the irreversible patients. Subsequent observations revealed that follistatin plays a key role in determining activin A signaling. 

Follistatin is a natural inhibitor of activin and is mainly synthesized in hepatocytes [19]. Physiologically, follistatin is released to suppress the follicle-stimulating hormone (FSH) [19]. The life history theory defines growth, reproduction, and maintenance as the three fundamental biological programs in humans [20]. In favorable environments, the synthesis and release of follistatin in hepatocytes is strictly regulated by the glucagon-to-insulin ratio to promote investment in growth and reproduction [19]. In harsh environments, follistatin is required for diversion of resources from reproduction to the defense arm [20]. Once MHN occurs, massive hepatocyte death not only releases large quantities of activin A, but also follistatin. This might benefit patients by temporarily sacrificing growth and reproduction function to direct more energy towards priority organs such as the brain, the heart, and the immune defense. However, the imbalance of the activin-to-follistatin ratio is “paid for” by dysregulated liver functions. If the balance of the activin-to-follistatin ratio is rapidly retrieved, LPC in the ALF patients possess active SMAD proteins and HNF4α expression, which maintain a normal international normalized ratio. In contrast, a lasting imbalance of the activin-to-follistatin ratio results in an impaired activin–HNF4α axis in LPC, and as a consequence poor survival in ALF patients [10].

Why can some patients rapidly rebalance the activin-to-follistatin ratio, but others cannot? In healthy persons, follistatin is regulated by insulin and glucagon: the former inhibits while the later induces follistatin expression and secretion in hepatocytes [21]. One unanswered question is the cellular source of follistatin in the condition of massive hepatocyte loss. To date, it is unknown whether LPC can also produce follistatin. In the condition of MHN, the remaining hepatocytes might be the main source of the hormone. As key systemic regulators, insulin and glucagon are critical to energy allocation in the disease syndromes relevant to systemic inflammatory response syndrome (SIRS), including ALF [20]. In SIRS, high levels of glucagon are required to maintain high levels of blood glucose [22]. To guarantee sufficient energy supply for priority organs and cells, insulin resistance occurs in major metabolic tissues such as adipose tissue, skeletal muscle, and hepatocytes [22]. Insulin resistance in hepatocytes compromises the inhibitory effect of insulin on follistatin and thus disrupts the ratio of glucagon-to-insulin, which controls follistatin synthesis. In irreversible ALF patients, most hepatocytes lose Glut2, indicating insulin resistance (unpublished data). This might explain why these patients have high levels of follistatin. Therefore, restoring insulin sensitivity in hepatocytes might be crucial for LPC function in ALF. This topic is beyond the range of the current review.

## 3. Monitoring Functional LPC and Systemic Hormones to Predict Clinical Outcome of ALF

Considering the crucial effects of HNF4α, activin A, follistatin, insulin and glucagon on the synthesis of coagulation factors, quantification of HNF4α and follistatin might provide predictive information regarding the progression and clinical outcome of ALF.

As a nuclear receptor and master hepatic transcription factor, detecting HNF4α requires histological examination. ALF is a severe disease syndrome, hence liver biopsy cannot be routinely performed in clinical practice. If by coincidence histological examination is available, evaluation of HNF4α expression provides a key parameter to reflect transcriptional regulation in the remaining hepatocytes and LPC, which largely determines the clinical outcome of ALF patients.

In contrast to HNF4α, activin A, follistatin, insulin and glucagon can be dynamically measured in patients’ blood. The role of insulin and glucagon in critical care diseases such as ALF has been elegantly described in previous reviews [22,23]. Our recent study describes a prospective clinical study investigating the association between serum follistatin levels and disease progression in cirrhotic patients [10]. Following 186 cirrhotic patients for 6 years, serum follistatin levels not only reflect mortality in ALF patients, but also the incidence of acute decompensation progressing to ACLF. As mentioned above, follistatin negatively regulates coagulation factor expression through inhibiting activin signaling. In addition, high levels of follistatin reflect an emergency condition that requires the host to reallocate energy resources towards maintenance by inhibiting reproduction [20]. Therefore, serum follistatin might be a reliable parameter to reflect emergency conditions in ALF patients.

## 4. Novel Therapeutic Approaches Based on the Activation of Liver Progenitor Cells

Given the critical and essential role of LPC in performing hepatocyte functions following MHN, therapies based on LPC activation represent a potential therapeutic lever for future ALF treatment. To date, stem cell therapy has been quite a hot field in clinical practice, including regarding liver disease [24]. The delivery of stem cells into patients’ bodies represents two challenges: (1) The allogeneic stem cells might be rejected by the host, and (2) how can we control the differentiation of stem cells into target cells, but not cancer cells? The performance of LPC in MHN suggests that the liver possesses its own stem cells, which are sufficient to take over key hepatocyte functions and differentiate into hepatocytes over time [6,10]. Whether LPC can handle the harsh disease conditions is dependent on the microenvironment. In irreversible ALF patients, LPC fail to take over hepatocyte functions. Therefore, modulating the disease microenvironment to activate LPC, the liver’s own stem cell reservoir, represents a new potential approach to rescuing ALF patients while circumventing the risk of external stem cell administration. 

## 5. Disease Model: A Bottleneck in Studying the Mechanisms of Acute Liver Failure

To date, the molecular mechanisms underlying LPC-derived liver regeneration in MHN-induced ALF remain largely unknown. A lack of suitable animal models is the major reason. Animal models of chronic liver damage, e.g., rodents fed with 3,5-diethoxycarbonyl-1,4-dihydrocollidine (DDC) or a choline-deficient ethionine-supplemented (CDE) diet, have provided some knowledge regarding the molecular mechanisms of LPC-mediated regeneration [6]. However, these models only mimic the disease scenario relevant to chronic liver disease, and do not match real ALF. Data from these models have led to a long-term controversial issue: whether or not LPC mediate liver regeneration in different disease conditions [9]. 

In different rodent models with chronic liver injury, the newly formed cells were mainly derived from neighboring hepatocytes, but not from oval cells, the rodent equivalent of LPC [25]. Only in mice fed with CDE for 4 months was a small fraction of regenerated hepatocytes derived from oval cells [25]. Recently, Forbes’ group reported that oval cell-derived hepatocytes, the rodent analogue of LPC-derived hepatocytes, can reach approximately 25% of newly generated cells when hepatocyte proliferation was inhibited by β1-integrin knockdown or p21 overexpression in liver-damaged mice [26]. Deng and colleagues observed that oval cell-derived hepatocytes accounted for 55.7 ± 3.9% in mice fed with thioacetamide (TAA) for 52 weeks and 23.3 ± 3.8% in those fed with DDC for 24 weeks [27]. These results suggest that LPC are not a predominant cell source for murine liver regeneration if the liver still possesses sufficient hepatocytes. An investigation based on ALF patients showed that 50% loss of hepatocytes is a threshold for extensive LPC activation [28]. We observed that even in ALF patients with MHN, there is no robust LPC activation in the areas with sufficient amounts of remaining hepatocytes [10]. To date, no rodent models can mimic massive hepatic necrosis and subsequent ALF because rodents do not receive intensive life support. Therefore, rodents cannot survive massive hepatic necrosis the same way that patients do. Hence, the insights obtained from rodent models do not explain the pathophysiological alteration in ALF.

Ductular reaction (DR), which is induced by LPC activation, is a key pathophysiological event in response to acute and chronic liver diseases. Desmet proposed four types of human DR in different disease circumstances [29]. DR type 1 results from proliferation of pre-existing cholangiocytes, resulting in elongation, branching and luminal widening of biliary tubes, which is predominant in acute complete bile duct obstruction, alpha-naphtyl isothiocyanate intoxication and cytokine-induced ductular increase. DR type 2A and 2B are both derived from ductular metaplasia or dedifferentiation of mature hepatocytes. The former is typically observed in periportal areas, most characteristically, though not exclusively, in chronic cholestatic conditions, while the latter occurs in parenchymal hypoxic areas: centrolobular in liver lobules and centronodular in cirrhotic nodules, in contrast to a predominantly periportal location of the other types. DR type 3 occurs in case of massive loss of parenchymal cells and consists of activation and proliferation of LPC located in ductules and canals of Hering. This is the pathophysiological alteration seen in MHN-induced ALF. The pathophysiological process in ALF can be briefly described as: (1) the liver receives etiologies (HBV or drug toxicity); (2) massive hepatic necrosis occurs; (3) systemic inflammatory response syndrome (SIRS) is initiated; (4) liver progenitor cells are activated; (5) whether liver progenitor cells take over hepatocyte functions, e.g., coagulation; and (6) the patient survives or receives liver transplantation. Unfortunately, no animal model encompassing all these pathophysiological features (e.g., massive hepatic necrosis, systemic inflammatory response syndrome, and LPC activation) is available to date. 

In contrast to rodents, zebrafish demonstrate robust cholangiocyte-derived liver regeneration when most hepatocytes are destroyed. Two elegant studies showed that administration of metronidazole killed nearly all hepatocytes of larval and adult zebrafish [30,31]. Once the toxin was washed out, the liver mass in these zebrafish was rapidly restored through cholangiocyte proliferation and transdifferentiation into hepatocyte-like cells [30,31]. These phenomena are very similar to those observed in humans. However, zebrafish do not possess Kupffer cells [32]. Hence, disease circumstance in zebrafish is not a suitable model either to investigate SIRS.

## 6. ALF, ACLF and Sepsis in Cirrhosis

Over the last decade, the emerging concept of acute-on-chronic liver failure (ACLF) illustrates different aspects of acute liver failure. On the other hand, the concept of ACLF itself faces challenge due to its frequently changed definition over time.

Acute-on-chronic liver failure, a term suggested by Jalan and Williams [33], emerged from studies showing the development of a syndrome associated with a high risk of short-term death (i.e., death < 28 days after hospital admission) in patients with acutely decompensated cirrhosis. This syndrome comprises three major features: (1) it occurs in the context of intense systemic inflammation; (2) it frequently develops in close temporal relationship with proinflammatory precipitating events (e.g., infections or alcoholic hepatitis); and (3) it is associated with single- or multiple-organ failure [34]. Although a substantial body of literature would recognize acute-on-chronic liver failure as a clinical entity, the existence of the syndrome is still controversially discussed [34]. Reviewing the history of the ACLF concept and its evolution, the following issues are noteworthy:

(1) To date, the mainstream ACLF definition is proposed by the European Association for the Study of the Liver–Chronic Liver Failure (EASL-CLIF) Consortium. This syndrome applies to patients with acutely decompensated cirrhosis, with or without prior decompensation, and does not exclude extrahepatic precipitating events [5]. Organ failure and severely dysregulated functions of the liver, kidney, and brain, as well as disturbed coagulation, circulation and respiration are the main standard to stratify patients into subgroups with different risks of death [5]. Different organs are considered to possess different importance in ACLF. Impressively, liver failure is less important than kidney failure to determine the clinical outcome of ACLF patients [5]. In the ACLF-establishment study, which enrolled 148 patients with ACLF grade 1, 108 with ACLF grade 2 and 47 with ACLF grade 3, liver failure was demonstrated in 37 (25.2%) patients with ACLF grade 1, 65 (60.2%) with ACLF grade 2, and 34 (63.8%) with ACLF grade 3, respectively [5]. This implies that liver failure is not essential to define this syndrome. Should a clinical syndrome in the absence of liver failure really be defined as acute-on-chronic “liver failure”? 

(2) Since the term ACLF was coined, its creators have been adopting the “Predisposition, Insult, Response, Organ Failure (PIRO)” scheme, a classical sepsis pathophysiology concept, to describe the disease process of ACLF [35,36,37]. If PIRO is the common pathophysiological feature in both sepsis and ACLF, how will one discriminate ACLF from sepsis in cirrhotic patients? In recent ACLF-related reviews, the EASL-ACLF experts seem to avoid the “PIRO” concept, rather asserting that “The pathophysiology of acute-on-chronic liver failure is still largely unknown” [34]. They stated that “Systemic inflammation may play a role” [34]. As we know, systemic inflammation is a prevailing feature of sepsis in cirrhosis [38]. Thus, these rhetoric alterations do not resolve the issue.

(3) One of the major arguments in ACLF is “proposed definitions of acute-on-chronic liver failure differ from one another” [34]. In contrast to the EASL definition of ACLF, APASL defined ACLF as “Acute hepatic insult manifesting as jaundice and coagulopathy, complicated within 4 weeks by ascites and/or encephalopathy in a patient with previously diagnosed or undiagnosed chronic liver disease” [39]. This definition is similar to the classic ALF definition except for the pre-existing disease prior to acute decompensation. For a long time, the ACLF debate between EASL and APASL was in part attributed to different patient populations between the West and the East. For example, alcoholic hepatitis and HCV related ACLF is the major etiology in populations across Europe and the USA, whereas chronic HBV infection is the dominant etiology in eastern countries except Japan [5,39]. It is noteworthy that both alcohol abuse and HCV infection do not result in MHN, whereas HBV infection is one of the leading factors leading to MHN [3]. In addition, alcoholic cirrhotic patients suffer from severely dysregulated microbiota and gut barrier dysfunction. When ACLF occurs in these patients, extrahepatic infection and subsequent systemic inflammation is a prevailing feature [40]. In contrast to ACLF caused by alcohol abuse and HCV infection, massive hepatocyte loss is the defining feature of HBV-ACLF [41]. Even in chronic HBV-infected patients, intrahepatic or extrahepatic ACLF shows different clinical outcomes [42], where the “battlefield” largely determines the clinical outcome of ACLF patients. If ACLF mainly reflects a disease occurring inside the liver, the APASL definition is suitable. If ACLF is only limited to the cirrhotic patients relevant to alcohol abuse and HCV infection, extrahepatic inflammation is the pathophysiological alteration pertinent to the disease [34]. However, how do we discriminate this type of ACLF from sepsis in cirrhosis? This question is still awaiting an answer.

## 7. Conclusions and Perspectives 

The activin–HNF4α–coagulation factor axis uncovers one aspect, be it central or just the so-called “tip of the iceberg”, of how a severely damaged liver can survive MHN. Massive hepatic necrosis is an extremely severe liver pathology leading to loss of most primary liver cells, which perform indispensable functions for the maintenance of systemic homeostasis. In such a severe disease condition, the liver suffers loss on three levels: structure, function, and regulation. As described by Medzhitov, any biological system/organ has three universal characteristics: “The elements of the system are arranged and interconnected into a particular structure, the structure supports a specific function, and regulation of the function is performed according to some logic of the system such as growth, stability, or coordination.” [43]. In the liver, hepatocytes are the cells performing primary functions, while other, local nonparenchymal cells such as Kupfer cells, liver sinusoidal endothelial cells, stellate cells and cholangiocytes, enable and support their performance [44]. In the condition of acute or chronic damage without massive hepatocyte loss, neighboring hepatocytes proliferate to recover lost parenchymal mass—the first pathway of liver regeneration in coordination with supporting cells [9]. In MHN, LPC are activated and take over primary cell functions—the second pathway of liver regeneration. The performance of LPC in MHN illustrates an efficient alternative mechanism of this key organ in response to an extremely severe disease. LPC are activated by massive hepatocyte loss regardless of the clinical outcome [6]. However, whether these activated LPC are capable of rapidly performing key hepatocyte functions such as coagulation is determined by the disease environment, including inflammation and the existence of etiology. In ALF, the first week after the onset of acute decompensation is a “golden window” that determines the survival of a patient [45]. Whether the disease environment is favorable for LPC to perform key functions within the critical clinical duration determines the prognosis of ALF. Therefore, the interaction between LPC and inflammatory cells should be intensively investigated. Figure 1 summarizes the state of the art regarding the sequence of pathophysiological alterations observed in ALF.

Several questions are open to future investigations: 

1. In a severe disease which kills most hepatocytes, how can LPC survive, proliferate and perform hepatocyte functions? 

2. The activin–HNF4α–coagulation factor axis in LPC determines coagulation function in ALF patients. How can LPC perform other key hepatocyte functions such as metabolism and de novo synthesis of glucose, albumin, bilirubin, and others? Are these processes under the control of HNF4α as well, or which other signals control these functions?

3. Dynamically altered local and systemic inflammation, including components, duration and magnitude play a crucial role in ameliorating or exacerbating ALF. Clarification of the impact of inflammation on LPC is a key issue in ALF.

4. How do systemic hormones such as insulin and glucagon influence inflammation and LPC performance?

## Figures and Tables

**Figure 1 biomolecules-12-00066-f001:**
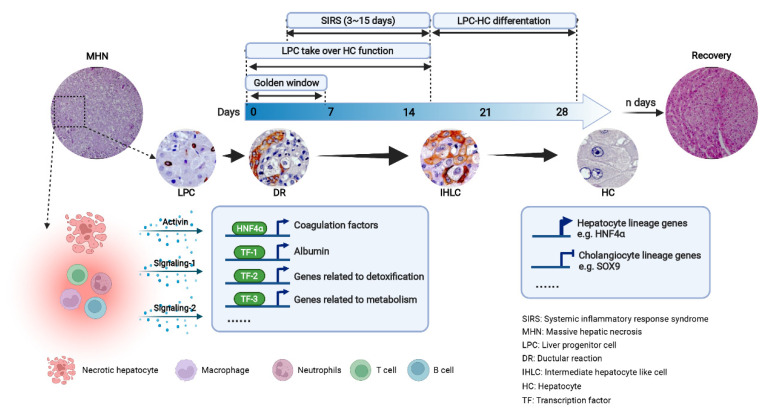
Activation of liver progenitor cells in massive hepatic necrosis-induced acute liver failure. Massive hepatic necrosis and subsequent severe inflammation activates liver progenitor cells (LPC). The first week since the onset of acute decompensation is a gold window for the survival of acute liver failure (ALF). Whether LPC can take over key hepatocyte functions, e.g., initiating the activin-HNF4α-coagulation factors axis and synthesis of albumin, among other proteins, determines clinical outcome of patients. Systemic inflammatory response syndrome occurs between day 3 and day 15 after disease initiation [12]. The interaction between the inflammatory environment and LPC is crucial for disease progression. After two weeks, intermediate hepatocyte-like cells (IHLC) emerge, suggesting that LPC begin to differentiate towards hepatocytes. Finalizing LPC-to-hepatocyte differentiation is characterized by inactivation of cholangiocyte lineage genes such as SOX9 and intensifying the expression of hepatocyte master genes such as HNF4α.

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
