# Peer review of "Liver Progenitor Cells in Massive Hepatic Necrosis—How Can a Patient Survive Acute Liver Failure?"

_biomolecules, 2022, doi:10.3390/biom12010066_

Round 1
Reviewer 1 Report
In this review, the authors focus on the mechanisms by which liver progenitor cells express HNF4α in massive hepatic necrosis, allowing the LPCs to produce coagulation factors, and thus promoting survival of patients with acute liver failure.
The work described in section 2.1 is interesting, but the nature of the alteration of the phenotype of LPCs that is described in that section is unclear. The authors refer to "LPCs" producing coagulation factors/taking over hepatocyte functions. Are these cells in fact still LPCs, or are they nascent hepatocytes derived from LPCs? It would be of interest to know what happens to these cells during the recovery phase from acute injury. Do they persist recognizably as LPCs, undergo apoptosis or all simply become hepatocytes?
In the conclusion of that section, the authors state that induction of HNF4α expression by LPCs explains why ALF patients are able to maintain "normal coagulation function in the condition of MHN." This does not make clinical sense, since the presence of coagulopathy is required for the diagnosis of ALF. Even patients who survive ALF will have abnormal coagulation at some point during their course.
While I agree with much of the authors' critique of the concept of ACLF (Section 6), this is quite far off the main topic of MHN. The topic is worthy of a commentary of its own, but it doesn't belong here.
Minor correction: at the end of line 7, page 4, the text reads "supporting prior" when what is intended is supporting priority.
Author Response
Dear reviewer
Please see our response in the attachment.

Reviewer 2 Report
This is a good review about liver progenitor cell (LPC) activation as potential mechanism that substitutes functions of necrotic hepatocytes in massive liver damage during acute liver failure (ALF). LPC regulation by the activin-HNF4α-coagulation factor axis represents new findings that may help detect the clinical outcome of massive hepatocyte loss-induced ALF. This review also describes involvement of systemic hormones (follistatin, insulin, and glucagon) in prediction of clinical outcome of ALF. Novel therapeutic approaches based on LPC and drawbacks in related preclinical disease models are also summarized. Finally, ALF, acute on chronic liver failure (ACLF) and sepsis in cirrhosis are also discussed. Unsolved questions are brought up for future investigations. Overall, this review is well written. There are some minor concerns.
(1) reference 10 is not completed. Journal that was cited but volume and page number were missing.
(2) Please describe EASL before its abbreviation in the first page of this paper.
Author Response

(The authors gave the same response as above.)

Reviewer 3 Report
This review is well and clearly written, covering a very important field of both fundamental and clinical research.
The topic is nicely reviewed, although mainly centered on the recent paper from the same group (Lin et al., Hepatology 2021).
I found the review instrcutive both at the clinical and biological levels, and I have no major comments.
Minor comments:
- page 3, paragraph 2.2, first line: "In normal livers, HNF4-a expression (please replace express) in LPCs is undetectable". Please indicate if LPCs by themselves could be easily detected in normal livers.
Author Response
Dear reviewer
Please see our response in the attchment.
